



# Technical note: An improved discharge sensitivity metric for young water fractions.

Francesc Gallart[1], Jana von Freyberg[2,3], María Valiente[4], James Kirchner[2,3], Pilar Llorens[1] and Jérôme Latron[1]

[1] Surface Hydrology and Erosion group, Department of Geosciences, IDAEA, CSIC, Barcelona, Spain
[2] Department of Environmental Systems Science, ETH Zurich, Zurich, Switzerland
[3] Swiss Federal Institute for Forest, Snow and Landscape Research (WSL), Birmensdorf, Switzerland
[4] Geodynamics Department, University of the Basque Country, Leioa, Spain

*Correspondence to*: Francesc Gallart (francesc.gallart@idaea.csic.es)

**Abstract:** Recent virtual and experimental investigations have shown that the young water fraction $F_{yw}$ (i.e. the proportion of catchment outflow younger than *circa* 2-3 months) increases with discharge in most catchments. The discharge sensitivity of $F_{yw}$ has been defined as the rate of increase in $F_{yw}$ with increasing discharge ($Q$), and has been estimated by the linear regression slope between $F_{yw}$ and $Q$, hereafter called $DS(Q)$. The combined use of both metrics, $F_{yw}$ and $DS(Q)$, provides a promising method for catchment inter-comparison studies that seek to understand streamflow generation processes. Here we

explore the discharge sensitivity of $F_{yw}$ in the intensively sampled small Mediterranean research catchment Can Vila. Intensive sampling of high flows at Can Vila allows young water fractions to be estimated for the far upper tail of the flow frequency distribution. These young water fractions converge toward 1 at the highest flows, illustrating a conceptual limitation in the linear regression method for estimating $DS(Q)$ as a metric of discharge sensitivity: $F_{yw}$ cannot grow with discharge indefinitely, since the fraction of young water in discharge can never be larger than 1. Here we propose to quantify

discharge sensitivity by the parameter of an exponential-type equation expressing how $F_{yw}$ varies with discharge. The exponential parameter ($S_d$) approximates $DS(Q)$ at moderate discharges where $F_{yw}$ is well below 1; however, the exponential equation and its discharge sensitivity metric better capture the non-linear relationship between $F_{yw}$ and $Q$ and are robust with respect to changes in the range of sampled discharges, allowing comparisons between catchments with strongly contrasting flow regimes.

**1 Recalling the definition of the discharge sensitivity of the young water fraction**

The seasonal cycles of stable isotopes in precipitation are damped and phase-shifted as they are transmitted through catchments, and thus can be used to infer properties of catchment travel-time distributions (e.g. DeWalle et al., 1997; McGuire and McDonnell, 2006). The young water fraction ($F_{yw}$), or the proportion of catchment outflow younger than *circa* 2-3 months, can be estimated as the ratio between the seasonal cycle amplitudes of stable water isotopes in precipitation and





stream water. This ratio consistently predicts $F_{yw}$ across a wide range of transit time distributions, whereas the same range of distributions yields widely varying mean transit times (Kirchner, 2016a).

The young water fraction usually increases with stream discharge (Kirchner, 2016b). To account for this flow-dependency in their study of 22 Swiss catchments, von Freyberg et al. (2018) distinguished between time-weighted ($F_{yw}$) and flow-weighted ($F^*_{yw}$) young water fractions and introduced the 'discharge sensitivity of the young water fraction' (which we term

$DS(Q)$) as a metric of the progressive increase of $F_{yw}$ with increasing catchment discharge ($Q$). Thus, by combining the mean $F^*_{yw}$ and its sensitivity to discharge, catchment young water response can be classified in two dimensions: catchments with low or high $F^*_{yw}$ and with low or high $DS(Q)$ (Figure 10 in von Freyberg et al., 2018). Because these two variables did not correlate with each other and correlated with different catchment characteristics, von Freyberg et al. (2018) suggested that $F^*_{yw}$ and $DS(Q)$ are two independent metrics that can be informative in catchment inter-comparison studies.

These authors used the linear slope between $F_{yw}$ (-) and discharge rate $Q$ (mm d$^{-1}$) for calculating $DS(Q)$ (d mm$^{-1}$). The use of discharge rate instead of volume rate (m$^3$ d$^{-1}$) is sensible, because of its independence from catchment area. They justified the choice of using $Q$ as forcing variable instead of log($Q$), which is more sensitive to low flows, by the main focus of the study being storm runoff generation.

Von Freyberg et al. (2018) determined $DS(Q)$ through a non-linear fitting algorithm. Assuming that the seasonal cycle

amplitude of stable water isotopes signal in stream water ($A_S$) varies with $Q$, but the corresponding cycle amplitude in precipitation ($A_P$) does not, then $F_{yw}$ varies with $Q$ as:

$$F_{yw}(Q) = A_S(Q)/A_P \qquad (1)$$

If $A_S$ is approximated as a linear function of $Q$,

$$A_S(Q) = n_S + m_S Q \qquad (2)$$

The slope ($m_S$) and the intercept ($n_S$) of this linear function can be obtained by fitting a sinusoid function to the seasonal variation of the isotopic signal of stream water $c_S(t)$ (‰):

$$c_S(t) = (n_S + m_S Q) \cdot sin(2\pi f t - \varphi_S) + k_S \qquad (3)$$

where $\varphi_S$ is the phase of the seasonal cycle (rad), $t$ is the time (decimal years), $f$ is the frequency (years$^{-1}$, equal to 1 for a full annual cycle) and $k_S$ (‰) is a constant describing the vertical offset of the isotope signal.

Combining Eqs. (1) and (2) yields:

$$F_{yw}(Q) = \frac{n_S}{A_P} + \frac{m_S}{A_P} Q \qquad (4)$$

Thus $DS(Q)$, the linear slope of the dependence of $F_{yw}$ on $Q$, can be approximated as $m_S/A_P$, which has units of $Q^{-1}$.



## 2 Investigating discharge sensitivity of the young water fraction in a small Mediterranean catchment

We applied the approach outlined above to the small Mediterranean Can Vila catchment (Vallcebre Research Catchments, Llorens *et al.*, 2018). The objectives were to better understand the Can Vila catchment's hydrology and to test the $F_{yw}$ and discharge sensitivity concepts and methods in an environment that was different, in terms of climate, catchment characteristics and sampling strategy, from the Swiss catchments studied by von Freyberg *et al.* (2018). This technical note focuses only on the aspects of this research that are relevant to the estimation of $F_{yw}$ and its discharge sensitivity, as other

aspects of the Can Vila catchment study will be presented in a separate publication (Gallart *et al.*, in preparation).

The Can Vila catchment is a 0.56 km$^2$, semi-humid (mean annual precipitation = 880 mm) Mediterranean mid-altitude (1,115-1,458 m asl) catchment with a rainfall-dominated flow regime. In addition to long-term hydrometric monitoring since the early 1990's, precipitation and stream water stable isotopes were sampled from May 2011 to September 2013 and from May 2015 to May 2016. During the isotope sampling period, 5-minute discharges ranged from zero to 2.621 m$^3$ s$^{-1}$ (4.68 m$^3$

s$^{-1}$ km$^{-2}$ or 404 mm d$^{-1}$), with a highly skewed flow duration curve (i.e., 30 % of stream flowed through the gauging station during 1 % of the time). A 'smart sampling strategy' was used to obtain flow-representative water samples, consisting of the combination of two automatic water samplers, one triggered by time and the other by flow. The resulting sampling intervals varied between 30 minutes and 26 days. We investigated the young water fraction and its discharge sensitivity for the Can Vila catchment by this 40 month-long isotope time series containing 464 precipitation and 858 streamflow samples. Given

the drier climate, the smaller catchment area and the much finer time scale for sampling, this data set extends the range of catchments investigated by von Freyberg *et al.* (2018).

For the Can Vila catchment, the flow-weighted young water fraction ($F^*_{yw}$=0.226±0.028) was much larger than the time-weighted young water fraction ($F_{yw}$=0.061±0.008). Both values fell within the range of those reported by von Freyberg *et al.* (2018), but the ratio between them was larger than at the Swiss catchments, suggesting that young water fractions are more

sensitive to discharge at Can Vila than at most of the Swiss sites.

To further explore the discharge sensitivity *DS(Q)* at Can Vila, we estimated young water fractions for different quantiles of the flow regime (similar to Figure 7 in von Freyberg *et al.*, 2018), extending the range to portray the highest flows (up to the top 0.25 %, (Figure 1). Our flow-dependent sampling strategy intensively sampled these high flows, which conventional sampling at regular time intervals would miss. Figure 1 shows that $F_{yw}$ increases with increasing discharge, from nearly 0 at

the lowest discharge to nearly 1 for $Q{\geq}24$ mm d$^{-1}$. This behaviour partly corresponds to a high-*DS(Q)* type 2 catchment in Figure 10 in von Freyberg *et al.* (2018). However, the non-linear behaviour of $F_{yw}$ with increasing flow shown in Figure 1 is inconsistent with a linear model of discharge sensitivity. Very small $F_{yw}$ values (<0.1) during baseflow are consistent with the long (7.7 years) mean transit time of base flows obtained in this catchment (Gallart *et al.*, 2016), whereas the high sensitivity of $F_{yw}$ to discharge reflects the varying pre-event water contributions (30-90 %) observed for different flow events

(Llorens *et al.*, 2018).



Equations (3) and (4) (numbered 9 and 10 in von Freyberg *et al.*, 2018) yield a discharge sensitivity $DS(Q)$ value of $0.0098\pm0.0012$ d mm$^{-1}$ for the Can Vila catchment (grey line in Figure 1), which is among the smallest discharge sensitivities obtained for the 22 Swiss catchments, in contrast with the visibly high discharge sensitivity of Can Vila over the range of its flow regime. Figure 1 shows that the linear design of $DS(Q)$ is clearly inadequate to capture the asymptotic

convergence of the young water fraction toward $F_{yw}\approx1$ at the far upper tail of the flow distribution. Highly dynamic catchments such as Can Vila, and flow sampling strategies like those employed here, demonstrate that a non-linear discharge sensitivity function is needed.

### 3 Defining alternative metrics for discharge sensitivity of the young water fraction

An alternative, non-linear model can be derived by noting that the sum of old and young water fractions is always 1, and by

assuming that the old water fraction decreases with increasing discharge and asymptotically approaches 0 (and thus the young water fraction asymptotically approaches 1) as $Q$ approaches infinity. We propose the following equation, where the old water fraction decreases exponentially with increasing $Q$, and the young water fraction grows accordingly:

$$F_{yw}(Q) = 1 - (1 - F_0) \cdot \exp(-Q \cdot S_d) \qquad (5)$$

where $F_0$ (-) is the $F_{yw}$ for virtual $Q=0$ and $S_d$ (unit of $Q^{-1}$) is the new discharge sensitivity metric. The red curve in Figure 1

shows the application of this equation to the Can Vila data.

On combining Eqs. (1), (3) and (5), the $F_0$ and $S_d$ parameters can be obtained by fitting a sinusoid function to the seasonal variation of the isotopic signal of stream water $c_S(t)$:

$$c_S(t) = A_P \cdot [1 - (1 - F_0) \cdot \exp(-Q(t) \cdot S_d)] \cdot sin(2\pi f t - \varphi_S) + k_S \qquad (6)$$

We obtained the $F_0$ and $S_d$ parameters with a non-linear analytic Gauss-Newton algorithm in which we used streamflow rates

as weights.

Taking the derivative of Eq. (5) with respect to $Q$ directly yields the result that the local discharge sensitivity $\frac{dF_{yw}(Q)}{dQ}$ at low discharges will be directly related to (and in many cases nearly equal to) $S_d$:

$$\begin{aligned}\frac{dF_{yw}(Q)}{dQ} &= (1 - F_0) \cdot S_d \cdot \exp(-Q \cdot S_d) \\ &\approx (1 - F_0) \cdot S_d \text{ for } Q \ll S_d^{-1} \\ &\approx S_d \text{ for } Q \ll S_d^{-1} \text{ and } F_0 \ll 1 \end{aligned} \qquad (7)$$

Because $F_0$ will typically be small, $S_d$ will typically be a good approximation to the slope of the relationship between $F_{yw}$ and

$Q$ at discharges that are low enough to keep $F_{yw}$ still far from 1.





**4 Sensitivity of the discharge sensitivity metrics to changes in data availability at the Can Vila catchment**

We used the Can Vila dataset to test the robustness of the $S_d$ metric, in comparison with the original $DS(Q)$ metric defined by von Freyberg *et al.* (2018) and with several alternative metrics designed to reduce or avoid some of the $DS(Q)$ metric's limitations. We investigated how these metrics changed when the discharge and water samples for determining $F_{yw}$ that

corresponded to the highest flows were excluded from the Can Vila dataset (Figure 2). This allowed us to test how these discharge sensitivity metrics were affected by the availability (or, conversely, the lack) of data encompassing extreme flows.

For this purpose, we compare the new $S_d$ metric, the original $DS(Q)$ metric and several dimensionless options that used $\log(Q)$, $Q/Q_{max}$, and $Q/Q_{mean}$ instead of $Q$ in the calculations ($Q_{max}$ and $Q_{mean}$ correspond to the maximum and mean values of the discharge rates $Q(t)$ associated with stream water sampling). We call the resulting discharge sensitivity metrics

$DS(logQ)$, $DS(Qmax)$ and $DS(Qmean)$, respectively. Note that $DS(Qmax)$ and $DS(Qmean)$ may be obtained by multiplying any previously calculated $DS(Q)$ value by $Q_{max}$ or $Q_{mean}$.

The new exponential $S_d$ metric values (Figure 2a) show some scatter but are robust to changes in the underlying data, exhibiting no systematic trend as the high flow observations were progressively discarded. In contrast, $DS(Q)$ is highly sensitive to changes in the analysed range of discharges (Figure 2b), rapidly increasing (by a factor of 5) on exclusion of the

highest flows from the calculations and reaching its maximum value on exclusion of the upper 5 % of flows ($Q$>4.82 mm d[-1]), corresponding to everything above the green dot (Top 5%) in Figure 1. Note that, as suggested by Eq. (7), $DS(Q)$ takes values similar to $S_d$ when the highest flows are excluded. $DS(logQ)$ declines promptly on omission of the highest flows (Figure 2c), but remains stable afterwards. $DS(Qmean)$ behaves similarly to $DS(Q)$, i.e. it is smallest when the complete data set is used and is largest on exclusion of the highest 5 % of flows from the analysis (Figure 2d). Finally, Figure 2e shows that

$DS(Qmax)$ becomes largest with the complete data set and sharply decreases to much smaller values on exclusion of the highest 1 % of flows from the calculations, but undergoes just a little progressive decrease when more data of the flow distribution are excluded.

In summary, $S_d$ is clearly more robust than the other discharge sensitivity metrics to changes in the sampled range of flows. It also has the distinct advantage that Eqs. (5)-(6), unlike Eqs. (3)-(4), can never yield $F_{yw}$ values larger than 1. One can see

from Eqs. (5)-(7) that $S_d$ functions as both a shape parameter, controlling how non-linear $F_{yw}$ is as it approaches 1, and a scale parameter, controlling the slope of the relationship between $F_{yw}$ and $Q$ at low or moderate discharges.

**5 Comparing discharge sensitivities at Can Vila and the Swiss catchments.**

Figure 3 compares the quantile plot of Figure 1 for the Can Vila catchment and the quantile plots of Figure 7 in von Freyberg *et al.* (2018) for the Swiss catchments of Langeten, Biber and Ilfis, which exhibit very different young water fractions and/or

discharge sensitivities (Table 1). The $F_0$ and $S_d$ metrics were calculated from Eq. (6) and good fits were obtained between the



individual $F_{yw}$ values and the median discharges as shown by the red curves. For comparison, grey curves correspond to the linear approach using eq. (4).

We find that young water fractions in the Can Vila catchment are among the most sensitive to discharge (largest $S_d$) between the Langeten and the Biber catchments. The young water fractions of the Ilfis catchment have almost no discharge sensitivity. Despite having the lowest $F_0$ value, which is in line with baseflow being several years old, the large discharge sensitivity value observed at Can Vila expresses well the highly dynamic streamflow regime in this Mediterranean mountain environment.

While the linear expression of discharge sensitivity (*DS(Q)*, Eq. 4) provides a reasonable fit for the low-to-medium flow regimes of the Swiss sites, it fails to capture the highly non-linear dependence of $F_{yw}$ on $Q$ at Can Vila, evidenced by the high flows sampled there (Figure 3a). In addition, Figure 3 shows a major drawback of the linear approach, which predicts $F_{yw}$ values larger than 1 for high-flow conditions.

The four catchments compared here differ considerably in catchment area and median discharge (Table 1), which often challenges a robust inter-comparison analysis. However, we show that Eq. (6) efficiently estimates the sensitivities ($S_d$) of $F_{yw}$ on $Q$ across these catchments.

The comparison of the $S_d$ and *DS(Q)* metrics for Can Vila and the 22 Swiss catchments studied by von Freyberg *et al.* (2018) demonstrates that the *DS(Q)* linear approach approximates small discharge sensitivities reasonably well (Figure 4). However, for relatively high discharge sensitivities, the linear model tends to predict smaller and more variable *DS(Q)* values. This behaviour may be attributed to the fact that, as shown in Figure 2b, when $S_d$ is high, the value of *DS(Q)* decreases if there are high flow samples that reduce the linear slope between $F_{yw}$ and $Q$ (as it happens in Figure 1).

In order to compare the frequencies of occurrence of $Q$ and $F_{yw}$ in the diverse catchments, the same points shown in Figure 3 are plotted on a single log-probabilistic graph in Figure 5. It presents the information as flow duration curves, using the corresponding quantile frequencies, the log-normal distributions fitted to the flow quantiles and the $F_{yw}(Q)$ lines obtained by applying Eq. (5) to the discharges. Figure 5 shows differences in behaviour between Can Vila and the three Swiss catchments due to the combination of flow distribution and discharge sensitivity of $F_{yw}$ that are only vaguely visible in Figure 3. This graph also allows anticipation of the $F_{yw}$ values that might be obtained if more samples would be collected during high flows (low exceedance frequencies) in the study catchments.

## Conclusions

The discharge sensitivity of the young water fraction is a promising metric for investigating streamflow generation processes and for catchment inter-comparison studies. However, the original linear regression approach between the young water fraction ($F_{yw}$) and discharge ($Q$) turns out to be inadequate when applied to an intensively sampled small catchment, because



it fails to predict $F_{yw}$ during high flows, which consist almost entirely of young water. The young water fractions converge toward 1 at the highest flows, revealing a conceptual limitation in the linear regression approach because $F_{yw}$ can never be larger than 1. Yet, as $F_{yw}$ is defined to range between 0 and 1, whereas $Q$ may vary by several orders of magnitude, the linear regression approach for estimating discharge sensitivity is sensitive to the highest $Q$ values at which $F_{yw}$ estimates are available, potentially hampers robust comparisons of discharge sensitivities at catchments with very different flow regimes.

We propose an alternative, exponential-type approach (Eq. 5) to overcome the limitations of the linear approach. The parameters of this exponential equation are $F_0$ (-), i.e., $F_{yw}$ for virtual zero discharge, and $S_d$ ($Q^{-1}$) that represents the shape of the curve for the increase of $F_{yw}$ with increasing $Q$. The exponential regression-based $S_d$ metric outperforms the linear regression-based discharge sensitivity metric in terms of physical soundness and lower sensitivity to changes in available tracer and discharge information.

As the proposed $S_d$ metric has dimensions inverse to discharge $Q$, its value depends on the units of $Q$ used in Eqs. (5) and (6). Nevertheless, the $S_d$ metric exhibited consistent behaviour across wide ranges of discharges sampled in the same catchment and between catchments of diverse sizes and flow regimes.

*Data availability*. The Swiss isotope data are available as detailed in von Freyberg *et al*. (2018). The Can Vila isotope data are available from Jérôme Latron upon request.

*Author contributions*. JL and PL designed the isotope sampling strategy at Can Vila and provided measurements. FG and MV analyzed the Can Vila data set. FG, JK and JF developed the new approach. FG prepared the paper with contributions from JF, JK, JL and PL.

*Competing interests*. The authors declare that they have no conflict of interest.

*Acknowledgements*. This research was supported by the projects TransHyMed (CGL2016-75957-R AEI/FEDER, UE) and Drought-CH (National Research Programme NRP 61 by the Swiss National Science Foundation). We are grateful to C. Cayuela, G. Bertran, M. Roig-Planasdemunt and E. Sánchez for their support during field work at the Can Vila catchment and to M. Eaude for his English style improvements.

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



**Table**

Table 1: Main characteristics and metrics of the catchments shown in Figures 2 and 3. $Q$ is the stream discharge, $F^*_{yw}$ is the flow-weighted young water fraction, $F_0$ is the young water fraction for virtual zero flow, and $S_d$ is the proposed discharge sensitivity
metric of the young water fraction, the last two defined in Eq. (5).

| Catchment | Area (km²) | median $Q$ (mm·d⁻¹) | coefficient of variation (%) | $F^*_{yw}$ (-) | $F_0$ (-) | $S_d$ (d·mm⁻¹) |
|---|---|---|---|---|---|---|
| Can Vila | 0.56 | 0.212 | 451.5 | 0.23 | 0.020±0.030 | 0.062±0.011 |
| Langeten | 60.3 | 1.49 | 61.7 | 0.07 | -0.043±0.034 | 0.070±0.017 |
| Biber | 31.6 | 1.54 | 149.4 | 0.39 | 0.170±0.059 | 0.058±0.013 |
| Ilfis | 187.9 | 1.74 | 113.6 | 0.12 | 0.110±0.025 | 0.003±0.005 |

**Figures**

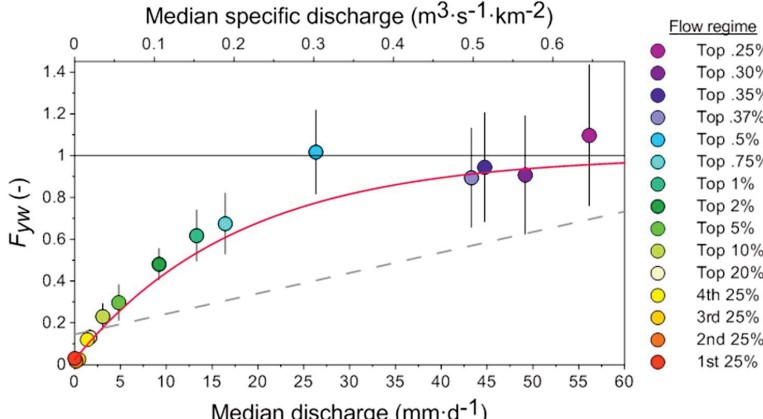

Figure 1: Variation in time-weighted young water fraction at the Can Vila catchment with increasing quantiles of the flow duration curve. The dashed grey line represents Eq. (4) and the red curve represents Eq. (5), using parameters obtained by fitting Eqs. (3) and (6), respectively, to all the stream water $\delta^{18}O$ isotope values. Maximum sampled discharge was 226 mm d⁻¹. Vertical bars represent standard errors.



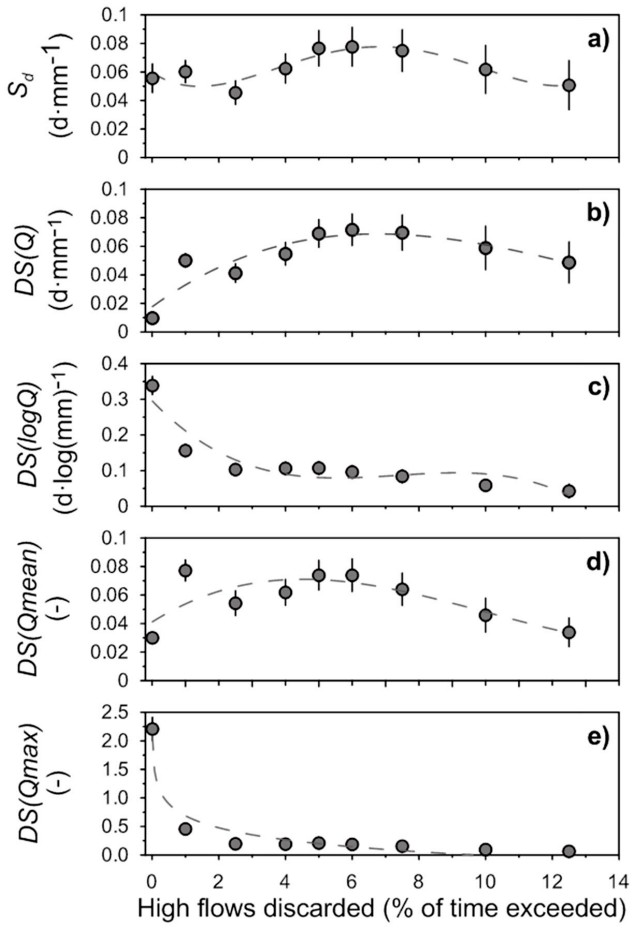


**Figure 2: Behaviour of the different discharge sensitivity metrics in the Can Vila catchment when measurements corresponding to the highest flows are sequentially discarded. Percentage of time exceeded refers to the flow duration curve. Vertical bars represent standard errors and dashed lines are ancillary polynomial fits.**





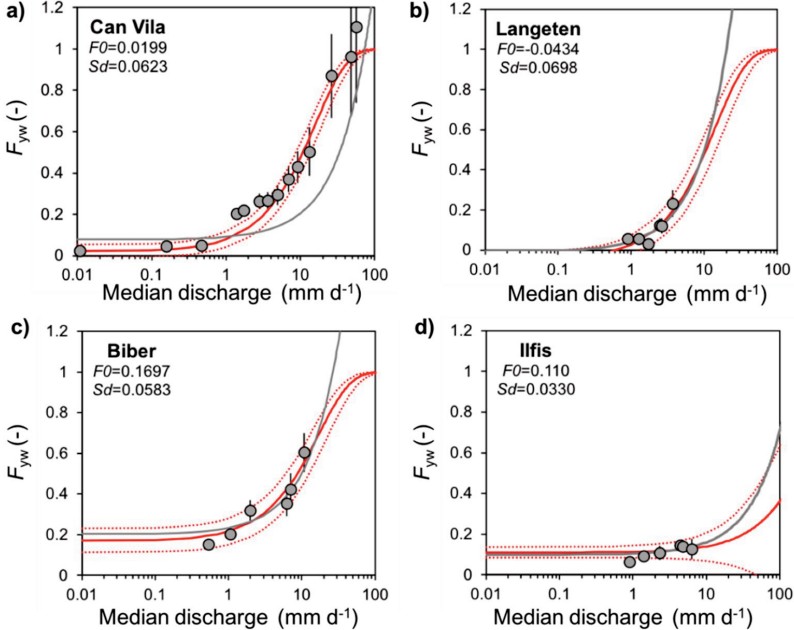


**Figure 3: Sensitivity of the young water fraction on discharge for the a) Can Vila, b) Langeten, c) Biber and d) Ilfis catchments. The red curves represent exponential fit (Eq. 5), with parameters $S_d$ and $F_0$ obtained through volume-weighted non-linear fitting of Eq. (6) to the stream water isotope data; red dashed lines indicate ±1 standard error. The grey curves represent the linear fit (Eq. 4).**


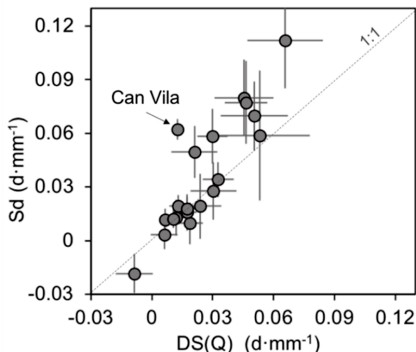

**Figure 4: Comparison of discharge sensitivities $DS(Q)$ and $S_d$ for 20 Swiss catchments and Can Vila (excluding Aach and Mentue for which unrealistic values for $DS(Q)$ or $S_d$ were obtained). Error bars indicate ±1 standard error.**





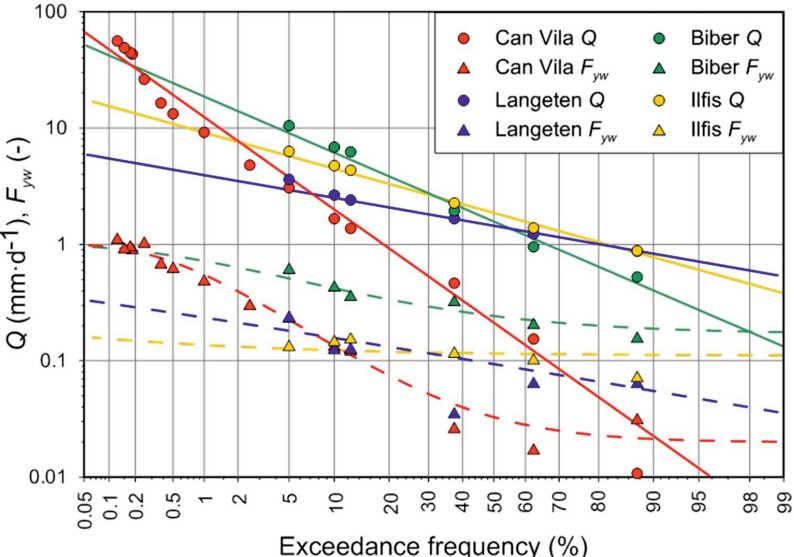


**Figure 5: Discharges and young water fractions from Figure 3 plotted against the respective quantile frequencies, along with the log-normal distributions fitted to discharges (continuous lines) and distributions of young water fractions (dashed curves) obtained by applying Eq. (5).**