# Peer review of "Technical note: An improved discharge sensitivity metric for young water fractions."

_Hydrology and Earth System Sciences, 2019_

## Referee Comment (RC1) · Anonymous Referee #1 · 4 Dec 2019

This paper explores the application of the young water fraction and discharge sensitivity of the young water fraction as a method for catchment comparison studies and understanding streamflow generation processes, following the method presented by von Freyberg et al., 2018. The authors show using data from their catchment that this method can be improved for high discharge events by using an exponential-type instead of linear equation to fit the relation between discharge and the young water fraction.

The manuscript is well written, easy to follow and of high quality. It is a nice contribution to the field of travel time distributions and young water fractions and the methodological advances are useful for other researcher using stable isotopes and young water fractions. There is not much to criticize, and the manuscript is almost publication-ready.

[Figure]

Below are some minor comments that should be addressed.

L66. What is the precipitation distribution in year? Would it be possible to show small graph of the annual precipitation and discharge? This would give readers an easy way to get a feeling of the study catchments compared to other catchments.

L70 mentions that sampling was done at maximum discharge of 404 mm/day. Caption of Figure 1 mentions that maximum sampled discharge was 226 mm/day.

L72. Was sampling done for rising and falling discharges? It would be interesting to see if there is hysteresis in the ywf and the high-frequent sampling might be suitable for this analysis.

Figure 1. If I understand correctly the 'Median discharge' in Figure 1 means the median discharge on the sampling day instead of the discharge at the exact moment of sampling? So all samples are in the figure, while the actual maximum/peak discharge at the moment of sampling on these days was 226 (or 404) mm/day. The authors could consider clarifying this.

L106. Also combining with Eqs 2. Or alternatively the authors could consider writing Eqs. (3) with 'AS(Q)' instead of '(nS + mSQ)', because when combining Eqs. 1, 3 and 5 'As' is not considered a linear function of Q, as in Eqs. 2.

L117-164. I compliment the authors on this analysis: it is very interesting to see what happens when highest flows are excluded in sampling and this helps in the comparison with the results of von Freyberg et al 2018. Looking at Figure 3 I wonder if the same results would be found if the lowest flows were excluded, e.g. <0.5 mm/d? It looks like a much better linear fit would then be reached, similar to the fits of von Freyberg et al 2018. Additionally, I am curious if their catchments have higher base flows or that the lowest flows were not sampled.

---

## Referee Comment (RC2) · Anonymous Referee #2 · 23 Dec 2019

In their technical note, Gallart et al. propose an improved discharge sensitivity metric for young water fractions. The manuscript develops original ideas/concepts and it is well structured and written.

Leveraging previous work by Freyberg et al. (2018), Gallart et al. first develop on the current state-of-the-art related to the potential for discharge sensitivity of the young water fraction to serve as a metric for investigating streamflow generation processes and catchment inter-comparison studies. The limitation of the original linear regression approach is clearly exposed - notably in the context of a catchment characterised by a rather flashy hydrological response to precipitation input. As a way forward, the authors propose the use of an exponential-type approach to overcome the limitation of the linear regression method.

[Figure]

More specifically, Gallart et al. leverage experimental data (hydro-meteorological & stable isotopes of O and H in precipitation and streamflow) sampled at high temporal resolution in their Mediterranean study catchment (Can Vila). Their streamflow sampling protocol was spanning a wide range of flow conditions (from low to high flows), providing a rare view into O and H stable isotope signature behaviour – almost along the entire range of the flow duration curve (FDC). Their dataset eventually allowed for the estimation of young water fractions also on the upper tail of the flow frequency distribution. Gallart et al. demonstrate that the conventional linear regression method used for estimating the discharge sensitivity suffers from a limitation in that Fyw cannot grow with discharge indefinitely. Ultimately, they introduce their novel discharge sensitivity metric, based on an exponential equation expressing how Fyw varies with discharge. Their approach clearly is a remarkable improvement, in that it clearly outperforms the linear regression-based discharge sensitivity metric – both in terms of its physical soundness and lower sensitivity to potential changes in available tracer and discharge information.

While the proposed solution is well documented, a few additional (minor) considerations may help to further sharpen the manuscript:

(i) When comparing catchments that are characterised by very contrasted climates (as it is the case here), it would be helpful to have more information on the hydro-meteorological context. For example, annual precipitation vs annual discharge, a flashiness index (e.g. as per Holko et al., 2011. doi:10.1016/j.jhydrol.2011.05.038) would equally be helpful in this context. A plot showing how stream water samples taken for isotope analysis are distributed along the FDC would also be very informative.

(ii) In equation 6, the parameters F0 and Sd are obtained via fitting a sinusoid function to the seasonal variation of the isotopic signal in stream water cs(t). In this context, it would be interesting to further investigate and discuss if and how the catchment's wetness state (changing across seasons, but also from one rainfall event to the next) may influence the hydrological functioning of the studied system – and subsequently

also the discharge sensitivity of the young water fraction. Soil moisture measurements (if available) or a (daily) water balance calculation could be helpful in this respect.

(iii) Along similar lines, are there any conclusions that can be drawn as to which reservoirs/compartments actually contribute to streamflow? Did the authors explore to what extent the intensity of precipitation events may influence hydrological responses – and trigger for example similar peak discharge for events that had different initial wetness states. Moderate rainfall may trigger high discharge when the catchment is already close to saturation; likewise, very intense precipitation may trigger similarly high discharge when the catchment has not yet reached saturation. In one case we may have saturation excess overland flow, as opposed to infiltration excess overland flow. How would this influence results and conclusions drawn on the discharge sensitivity of the young water fraction? How much would this also impact any potential catchment intercomparison between catchments with contrasted climate characteristics?

(iv) One of the main conclusions of the manuscript is that there is a need for sampling intensively the largest possible range of discharge values along the flow duration curve – with a special focus on (very) high flows. Considering potential hysteretic patterns in the rating curves, how would they impact the sampling protocol and subsequently the conclusions drawn from the obtained data? Is the dataset available for the Can Vila catchment (spanning a wide range of discharge values for O and H stable isotopes in stream water) offering the possibility to investigate this question?

(v) Possibly, the authors could conclude their work by stating one or two hypotheses that they may consider as being important to be tested in future work (for example in other physiographic contexts).
* * *

---

## Author Comment (AC1) · 15 Jan 2020

Responses to referee #1

We thank this referee for the fair and useful comments she/he made. We have addressed all comments for improving the clarity and soundness of the paper.

L66. What is the precipitation distribution in year? Would it be possible to show small graph of the annual precipitation and discharge? This would give readers an easy way to get a feeling of the study catchments compared to other catchments.

Response: Because our manuscript is a short technical note that does not focus on a scientific analysis of Can Vila's hydrological properties, in response to this comment we decided to include a sentence on the precipitation and runoff regimes of the Can

Vila catchment along with some additional hydro-climatic indices in Table 1. Furthermore, two publications in which Can Vila's climatic and hydrologic properties have been described previously will be cited.

L70. mentions that sampling was done at maximum discharge of 404 mm/day. Caption of Figure 1 mentions that maximum sampled discharge was 226 mm/day.

Response: The maximum recorded discharge was 404 mm/day. However, the maximum discharge that was actually sampled was substantially smaller (226 mm/day). This will be better explained in the revised manuscript to avoid any misunderstanding.

L72. Was sampling done for rising and falling discharges? It would be interesting to see if there is hysteresis in the ywf and the high-frequent sampling might be suitable for this analysis.

Response: At Can Vila, the rate of sampling was higher during the rising limb of the hydrograph than during the falling limb because the discharge increase was much faster during the first. This technical detail will be explained better in the revised text. The possible hysteresis is an interesting question not yet investigated.

Figure 1. If I understand correctly the 'Median discharge' in Figure 1 means the median discharge on the sampling day instead of the discharge at the exact moment of sampling? So all samples are in the figure, while the actual maximum/peak discharge at the moment of sampling on these days was 226 (or 404) mm/day. The authors could consider clarifying this.

Response: Discharges were measured at the time of sampling and were not aggregated but simply transformed into daily flow units for easier comparison with other studies. For our analysis in Fig. 1, we only used those discharge values (in mm/day), during which water samples were collected. Then, the "Median discharge" in Figure 1 is the median value of sampled discharges for each flow regime (e.g., 1st 25%). This will be better explained to avoid this misunderstanding.
L106. Also combining with Eqs 2. Or alternatively the authors could consider writing Eqs. (3) with 'AS(Q)' instead of '(nS + mSQ)', because when combining Eqs. 1, 3 and 5 'As' is not considered a linear function of Q, as in Eqs. 2.

Response: Following the reviewer's suggestion we will include the general expression for Cs after Kirchner (2016):

$Cs(t) = As \sin(2 \pi f t - \varphi s) + ks$ Eq.(2rev)

before Eq. (2) in the revised manuscript. Eq. (3) follows from inserting Eq. (2) into Eq. (2rev). In section 3, we will explain better how Eq. (6) was obtained: Eq. (5) was proposed based on the search for an exponential function to describe the data points in Fig. 1. We combined Eq. (5) with Eq. (1), and re-arranged the formula so that only As(Q) remains on the left side of the equation:

$As(Q) = Ap [1-(1-Fo) \exp(-Q(t) Sd)]$ Eq.(6rev)

By inserting Eq. (6rev) into Eq. (2rev), we obtain Eq. (6), which allows for estimating Sd and Fo from Cs.

L117-164. I compliment the authors on this analysis: it is very interesting to see what happens when highest flows are excluded in sampling and this helps in the comparison with the results of von Freyberg et al 2018. Looking at Figure 3 I wonder if the same results would be found if the lowest flows were excluded, e.g. <0.5 mm/d? It looks like a much better linear fit would then be reached, similar to the fits of von Freyberg et al 2018. Additionally, I am curious if their catchments have higher base flows or that the lowest flows were not sampled.

Response: At Can Vila, excluding the lowest flows does not change the results in Figure 2, unless DS(logQ) is used. 0.5 mm/day are exceeded only 32.4% of the time (see red line in Figure (5)), so cannot be considered low flows there. Low flows were also sampled at the Swiss catchments, although they were much higher (see also Figure (5)).

Kirchner, J. W.: Aggregation in environmental systems – Part 1: Seasonal tracer cycles quantify young water fractions, but not mean transit times, in spatially heterogeneous catchments, Hydrol. Earth Syst. Sci., 20, 279–297, https://doi.org/10.5194/hess-20-279-2016, 2016.

---

## Author Comment (AC2) · 15 Jan 2020

We thank the reviewer for the comments and suggestions she/he made. Most of the reviewer's suggestions focus on how the concept of young water fraction and its discharge sensitivity can be applied to analyse the hydrological functioning of the catchments; however, such analyses would go beyond the scope of a technical note. Nevertheless, we will respond to all comments below and make the opportune changes in the manuscript.

(i) When comparing catchments that are characterised by very contrasted climates (as it is the case here), it would be helpful to have more information on the hydrometeorological context. For example, annual precipitation vs annual discharge, a flashiness

[Figure]

index (e.g. as per Holko et al., 2011. doi:10.1016/j.jhydrol.2011.05.038) would equally be helpful in this context. A plot showing how stream water samples taken for isotope analysis are distributed along the FDC would also be very informative.

Response: Following this suggestion, several more hydro-climatic indices will be included in Table 1 in the revised manuscript for easier comparison with the catchments studied in von Freyberg et al. (2018) and elsewhere: Annual precipitation and discharge, average precipitation intensity and quick flow index. At Can Vila, where the streamwater sampling frequency increased as a function of discharge, more information on the time exceedance of recorded and sampled discharges will be added. More information about the hydro-climatic properties of the Can Vila catchment is available elsewhere and the relevant publications will be cited.

(ii) In equation 6, the parameters Fo and Sd are obtained via fitting a sinusoid function to the seasonal variation of the isotopic signal in stream water Cs(t). In this context, it would be interesting to further investigate and discuss if and how the catchment's wetness state (changing across seasons, but also from one rainfall event to the next) may influence the hydrological functioning of the studied system – and subsequently also the discharge sensitivity of the young water fraction. Soil moisture measurements (if available) or a (daily) water balance calculation could be helpful in this respect.

Response: We agree that these comments would be relevant for analysing the hydrological functioning of the catchment, which is, however, not the scope of our technical note. The technical note is intended to improve a metric primarily designed for describing the time-aggregated response of the catchments.

(iii) Along similar lines, are there any conclusions that can be drawn as to which reservoirs/ compartments actually contribute to streamflow? Did the authors explore to what extent the intensity of precipitation events may influence hydrological responses – and trigger for example similar peak discharge for events that had different initial wetness states. Moderate rainfall may trigger high discharge when the catchment is already

close to saturation; likewise, very intense precipitation may trigger similarly high discharge when the catchment has not yet reached saturation. In one case we may have saturation excess overland flow, as opposed to infiltration excess overland flow. How would this influence results and conclusions drawn on the discharge sensitivity of the young water fraction? How much would this also impact any potential catchment intercomparison between catchments with contrasted climate characteristics?

Response: These comments are relevant for extending the application of the young water fraction concept to the analysis of the hydrological functioning of catchments. For our technical note, we utilized the Can Vila data with the intention to demonstrate the limitations of a linear discharge sensitivity of Fyw and to develop an alternative approach. An analysis of the hydrological processes that are responsible for the observed discharge sensitivity in Can Vila would go beyond the scope of the technical note.

(iv) One of the main conclusions of the manuscript is that there is a need for sampling intensively the largest possible range of discharge values along the flow duration curve – with a special focus on (very) high flows. Considering potential hysteretic patterns in the rating curves, how would they impact the sampling protocol and subsequently the conclusions drawn from the obtained data? Is the dataset available for the Can Vila catchment (spanning a wide range of discharge values for O and H stable isotopes in stream water) offering the possibility to investigate this question?

Response: Indeed, the need for high-frequency stream water sampling is an indirect conclusion of our analysis, more widely analysed in Gallart et al. (in review). But this issue is partly offset by the fact that the new exponential Sd metric is much less sensitive to the largest sampled discharges than the original linear DS(Q). At Can Vila, the rate of sampling was higher during the rising limb of the hydrograph than during the falling limb because the discharge increase was much faster than during recession. This opens the possibility to investigate the potential hysteresis in the rating curve, an interesting question not yet attempted.

(v) Possibly, the authors could conclude their work by stating one or two hypotheses that they may consider as being important to be tested in future work (for example in other physiographic contexts).

Response: Following this comment we will include some open questions directly related to the role of the sampling design in robustly determining Sd and Fyw, as well as whether Sd, F0 and Fyw are correlated with each other when diverse catchments are compared.

Gallart, F., Valiente, M., Llorens, P., Cayuela, C., Sprenger, M., Latron,J.: Investigating young water fractions in different hydrological compartments of a small Mediterranean mountain catchment: both precipitation forcing and sampling frequency matter. Hydrol. Process. (in review)